# Prevalence of Sarcopenia and Its Impact on Cardiovascular Events and Mortality among Dialysis Patients: A Systematic Review and Meta-Analysis

**DOI:** 10.3390/nu14194077

**Published:** 2022-09-30

**Authors:** Wannasit Wathanavasin, Athiphat Banjongjit, Yingyos Avihingsanon, Kearkiat Praditpornsilpa, Kriang Tungsanga, Somchai Eiam-Ong, Paweena Susantitaphong

**Affiliations:** 1Nephrology Unit, Department of Medicine, Charoenkrung Pracharak Hospital, Bangkok Metropolitan Administration, Bangkok 10120, Thailand; 2Division of Nephrology, Department of Medicine, Faculty of Medicine, Chulalongkorn University, Bangkok 10120, Thailand; 3Research Unit for Metabolic Bone Disease in CKD Patients, Faculty of Medicine, Chulalongkorn University, Bangkok 10120, Thailand

**Keywords:** sarcopenia, hemodialysis, peritoneal dialysis, end-stage kidney disease, prevalence, mortality

## Abstract

Sarcopenia in end-stage kidney disease patients requiring dialysis is a frequent complication but remains an under-recognized problem. This meta-analysis was conducted to determine the prevalence of sarcopenia and explored its impacts on clinical outcomes, especially cardiovascular events, and mortality in dialysis patients. The eligible studies were searched from PubMed, Scopus, and Cochrane Central Register of Controlled trials up to 31 March 2022. We included studies that reported the interested outcomes, and the random-effects model was used for analysis. Forty-one studies with 7576 patients were included. The pooled prevalence of sarcopenia in dialysis patients was 25.6% (95% CI 22.1 to 29.4%). Sarcopenia was significantly associated with higher mortality risk (adjusted OR 1.83 (95% CI 1.40 to 2.39)) and cardiovascular events (adjusted OR 3.80 (95% CI 1.79 to 8.09)). Additionally, both low muscle mass and low muscle strength were independently related to increased mortality risk in dialysis patients (OR 1.71; 95% CI (1.20 to 2.44), OR 2.15 (95% CI 1.51 to 3.07)), respectively. This meta-analysis revealed that sarcopenia was highly prevalent among dialysis patients and shown to be an important predictor of cardiovascular events and mortality. Future intervention research to alleviate this disease burden in dialysis patients is needed.

## 1. Introduction

Sarcopenia, a disorder defined as having a reduction in skeletal muscle mass with declining muscle strength and function [1], is a frequent complication in patients with chronic kidney disease (CKD), especially in end-stage kidney disease (ESKD) patients undergoing kidney replacement therapy (KRT). It has been shown to be an important predictor of patients’ falls and fractures, mobility disorders, dependency, low quality of life, and death [2,3]. Interest in sarcopenia increased after the European Working Group for Sarcopenia for Older People (EWGSOP) established the Sarcopenia Consensus in 2010 [4]. There were exponentially increased publications of this disorder in various populations [5], including chronic dialysis patients [6,7]. Moreover, many recent studies have indicated that there is an observed association between sarcopenia and various clinical outcomes.

Beyond the aging-related process, ESKD-related sarcopenia was categorized as secondary sarcopenia due to several vulnerable factors, including the accumulation of uremic toxins, metabolic acidosis, malnutrition, loss of amino acids in dialysis therapy, and typical low-grade chronic inflammatory status [6,7]. These elements together result in a final process of increasing protein degradation and reducing protein synthesis, leading to a negative nitrogen balance [8,9]. Finally, dialysis patients tend to have low physical activity, resulting in aggravation of muscle loss [10].

Indeed, studies regarding the prevalence of sarcopenia among dialysis patients yield widely varying results, depending on the populations, the methods used to measure muscle mass, and the diagnosis criteria [2,4,10,11,12,13,14,15]. As a result of the large discrepancies in definition, sarcopenia prevalence in patients undergoing KRT is, to date, poorly defined. It remains an under-recognized problem that may hinder the early intervention, leading to poor clinical outcome. This systematic review and meta-analysis aimed to determine the global prevalence of sarcopenia in patients with ESKD undergoing both hemodialysis (HD) and peritoneal dialysis (PD) modalities and explore whether it has significant association with the clinical outcomes, especially cardiovascular (CV) events and mortality. Such information may give growing attention to sarcopenia and enhance future intervention research to address effective clinical service programs in dialysis patients.

## 2. Methods

We performed this systematic review and meta-analysis following the Preferred Reporting Items for Systematic Reviews and Meta-Analyses (PRISMA) 2020 [16] guidelines for reporting systematic reviews of interventions and a prespecified registered protocol in the PROSPERO database (registration number CRD 42022324618).

### 2.1. Searching Strategy

We conducted a systematic search of relevant articles in the database PubMed incorporating Medical Subject Heading Indexation (MeSH) terms, Scopus, and Cochrane Central Register of Controlled trials up to 31 March 2022. Keywords were detailed in Appendix A. The search had no language restriction and focused on adults (age ≥ 18 years).

### 2.2. Study Outcomes

The primary outcome was the prevalence of sarcopenia in dialysis patients. We decided to focus on the prevalence estimates derived from validated and well-established diagnostic criteria widely used internationally, including the European Working Group on Sarcopenia in Older People (EWGSOP), Asian Working Group for Sarcopenia (AWGS), Foundation for the National Institutes of Health Sarcopenia Projects (FNIH), and International Working Group on Sarcopenia (IWGS). We also planned to explore the association of sarcopenia with clinical outcomes. The clinical outcomes of interest were CV events (defined as any event related to the cardiovascular system including myocardial infarction, stroke, hypertensive crisis, heart failure, or unspecified cardiovascular serious adverse events reported according to the definition of each trial) and all-cause mortality.

### 2.3. Eligibility Criteria

Studies that were eligible for inclusion were: (1) the population consisting of adults (≥18 years) with ESKD on dialysis (HD or PD); (2) the prevalence of sarcopenia was reported; (3) sarcopenia was defined by assessing both low muscle mass (LMM) and low muscle strength (LMS); (4) the study reported baseline characteristics of both sarcopenic and non-sarcopenic patients. Exclusion criteria were as follows: (1) the study was an animal study, a review article, case report, or case series; (2) the study focused only on a specific group of sarcopenias (e.g., sarcopenic obesity); (3) the study did not stratify dialysis patients from patients with other spectra of kidney diseases: non-dialysis-dependent CKD, and patients undergoing kidney transplantation.

### 2.4. Study Selection

According to the eligibility criteria, two authors (AB and WW) separately screened the title and abstract of each retrieved record. Then, AB and WW independently reviewed and rescreened the full text of selected reports. A third author (PS) resolved the disagreement between the first two reviewers by discussion. The largest sample size report was selected if the reports resulted from the same cohort.

### 2.5. Data Extraction and Risk of Bias Assessment

Two authors (AB and WW) independently carried out data extraction. The data tables were categorized in topics, including author, year of publication, study design, number of the study population, baseline characteristics of patients (age, sex, dialysis vintage, mode of KRT), operational criteria for the diagnosis of sarcopenia, measurement tools (for assessing muscle mass, muscle strength and/or physical performance), duration of follow-up, the prevalence of sarcopenia as well as CV events, and mortality rate in cohort studies.

The risk of bias was assessed using the Newcastle–Ottawa Scale (NOS) for cohort studies, and the NOS was adapted for cross-sectional studies [17,18]. The NOS includes a series of questions used to assess the choice of study participants, the comparability of the population, and the determination of exposure or outcomes. The NOS for cohort studies was converted to AHRQ standards as good, fair, or poor quality. The NOS was adapted for cross-sectional studies with a maximum score of 10 points. The 9–10 points study was categorized as very good studies; 7–8 points as good studies; 5–6 points as satisfactory studies and 0–4 points as unsatisfactory studies. Using these checklists, two authors (AB and WW) evaluated each of the included articles for their quality. Divergent views were resolved by consultation with a third author (PS) (Appendix A).

### 2.6. Statistical Analysis

The results of the systematic review were qualitatively tabulated and synthesized. For a subset of studies with analyzable and comparable data, the results were synthesized quantitatively by performing random-effects model meta-analyses to compute absolute net differences in continuous variables (i.e., age, body mass index (BMI), serum albumin, phosphate, parathyroid hormone (PTH), creatinine (Cr), C-reactive protein (CRP), 25-OH vitamin D, hemoglobin (Hb), and Kt/V) and pooled odd ratios (OR) for binary variables (i.e., presence versus absence of sarcopenia) from univariate and multivariate analyses. The data from both analyses were retrieved for meta-analysis. All pooled estimates were displayed with a percentage and 95% confidence interval (CI). Existence of heterogeneity among effect sizes of individual studies was determined by using the Cochrane’s Q test and the I^2^ index, with a value of 75% or greater indicating medium-to-high heterogeneity. To explore sources of heterogeneity, we performed subgroup meta-analyses according to KRT modalities (HD or PD), race (Asian or non-Asian), time of muscle mass measurement (pre-HD, intra-HD, or post-HD), the tool of muscle mass measurement (dual-energy X-ray absorptiometry (DXA), bioelectrical impedance spectroscopy (BIS), bioelectrical impedance analysis (BIA) or calf circumference (CC)), follow-up duration (less or more than 2 years), and components of diagnostic criteria (LMM or LMS). Forest plot was used as a graphical representation of heterogeneity among the included studies. Publication bias was formally evaluated by Funnel plots and the Egger test. The analyses were performed using Comprehensive Meta-Analysis version 2.0 (www.meta-analysis.com (accessed on 4 August 2022); Biostat, Englewood, NJ, USA).

## 3. Results

### 3.1. Characteristics of the Included Studies

A total of 491 potentially relevant records were initially identified from the database search. After removing 61 duplicated records, 430 record titles and abstracts were screened according to the inclusion and exclusion criteria resulting in 55 full text publications that were further screened. After full-text screening, 14 reports were excluded (the reasons for exclusion are detailed in Figure 1). Finally, 41 studies fulfilled the inclusion criteria and were included in the systematic review and meta-analysis (Figure 1).

The characteristics of the individual study are shown in Table 1. There were 25 cross-sectional studies and 16 cohort studies with a total of 7576 patients. The studies were published between 2014 and 2022 and varied in sample size (33 to 645 patients). Among the included studies, twenty-one studies were conducted in Asian countries (China, Iran, Japan, Korea, Taiwan, Thailand, and Turkey) [19,20,21,22,23,24,25,26,27,28,29,30,31,32,33,34,35,36,37,38,39], nine in European countries (France, Sweden, Spain, and United Kingdom) [40,41,42,43,44,45,46,47,48], eight in South American countries (Brazil and Argentina) [49,50,51,52,53,54,55,56], two in the USA [57,58], and the remaining one in Australia [59]. These studies yielded a total sample of five continents (Asia: 45.6%, Europe: 25.9%, North America: 14.3%, South America: 13.7%, and Australia: 0.5%). Thirty-one studies [19,22,23,24,25,26,27,29,30,34,35,36,37,38,39,40,41,43,44,46,48,49,50,51,52,53,54,55,56,57,58] were conducted in HD population, seven studies [20,21,28,31,32,42,47] were performed in PD population, and three studies [24,33,59] were carried out in mixed populations (both HD and PD). Of the included patients, the mean age was 62.3 years, 61.4% were male, and 38.6% had diabetes mellitus as a comorbidity with a mean dialysis vintage of 52.4 months.

### 3.2. Operational Criteria for the Diagnosis of Sarcopenia

The diagnostic methods and measurement tools for muscle assessment of the individual study are shown in Table 1. All studies assessed muscle mass plus muscle strength and/or physical performance to fulfill the validated operational diagnostic criteria of sarcopenia. The EWGSOP 2010 criteria (*n* = 17:41.5%) was the most utilized. Regarding measurement tools for assessing muscle mass, BIA (*n* = 26:63.4%) and DXA (*n* = 10:24.4%) were the two mainly selected. All muscle strength was assessed by handgrip test (*n* = 41:100%). Physical performance was evaluated in nearly half of all studies (*n* = 19:46.3%) mainly by gait speed test.

### 3.3. Methodological Quality

Regarding the Newcastle–Ottawa Scale (NOS) for cohort studies and the NOS adapted for cross-sectional studies [17,18], all cohort studies (*n* = 16:100%) were considered as good quality. Most of the cross-sectional studies (*n* = 24:96%) were considered as good quality (score of 7–8), while the remaining study (*n* = 1:4%) was considered as satisfactory quality (score of 5–6) (see Appendix A).

### 3.4. Prevalence of Sarcopenia in Dialysis Patients

#### 3.4.1. The Overall Pooled Prevalence of Sarcopenia in Dialysis Patients

The prevalence of sarcopenia in dialysis patients ranged from 1.5% to 68%, and the overall pooled prevalence of sarcopenia was 25.6% (95% CI: 22.1% to 29.4%). There was substantial heterogeneity among the studies (I^2^ = 91.98%, *p* < 0.001, Figure 2).

#### 3.4.2. Subgroup Analysis

Table 2 summarizes the subgroup analyses examining the pooled prevalence of sarcopenia in different variables among dialysis patients. Among various diagnostic criteria, the highest prevalence was identified by the AWGS 2019 criteria (36.9%, 95% CI 30.4% to 44.2%). The others, IWGS, EWGSOP 2010, EWGSOP 2019, AWGS 2014, and FNIH were 34.9%, 24.4%, 24.1%, 22%, and 20%, respectively. Assessed by continents worldwide, the lowest absolute prevalence was observed in North America (15.4%), increasing through Australia (17.9%), South America (20.4%), Asia (27.9%), and reaching the highest levels in Europe (29.1%) (Appendix A).

In term of dialysis modality, the prevalence of sarcopenia was significantly higher in the HD population compared with the PD population (26.8%, 95% CI 22.8% to 31.2%, and 17.5%, 95% CI 11.9% to 24.8%, p for interaction = 0.037), respectively. Moreover, the timing of measurement was also important; the prevalence of sarcopenia was slightly lower but not significantly when measured at pre-dialysis, compared with post-dialysis assessment (21.5%, 95% CI 15.0% to 29.8%, and 27.8%, 95% CI 22.2% to 34.3%), respectively (*p* for interaction = 0.21).

### 3.5. Meta-Analysis of Baseline Characteristics between Patients with or without Sarcopenia

Men and diabetes mellitus (DM) patients seemed to have higher risk of sarcopenia ((OR 1.06, 95% CI 0.72 to 1.57; 21 studies, 5440 patients, *p* = 0.76) and OR 1.20, 95% CI 0.72 to 1.57; 18 studies, 3319 patients, *p* = 0.18)), respectively. Dialysis patients with sarcopenia were significantly associated with older and longer dialysis vintage ((8.8 years, 95% CI 7.1 to 10.5, *p* < 0.001, I^2^ = 73.80%) and 5.6 years, 95% CI 0.9 to 10.2, *p* = 0.02, I^2^ = 10.50%, respectively)), higher CRP, and higher dialysis adequacy by Kt/V were also observed. (1.3 mg/dL, 95% CI 0.07 to 2.54, *p* = 0.038, I^2^ = 92.01%) and Kt/V 0.11, 95% CI 0.06 to 0.17, *p* < 0.001, I^2^ =71.09%, respectively) (Table 3).

Dialysis patients with sarcopenia had significantly lower serum albumin, serum phosphate, serum parathyroid hormone, serum creatinine, and serum 25-OH vitamin D ((−0.1 g/dL, 95% CI −0.2 to −0.1, *p* < 0.001, I^2^ 54.15%), (−0.6 mg/dL, 95% CI −0.8 to −0.4, *p* < 0.001, I^2^ 43.06%),(−43.4 pg/mL, 95% CI −94.6 to −2.2, *p* = 0.04, I^2^ 52.89%), (−1.6 mg/dL, 95% CI −2.0 to −1.3, *p* = 0.05, I^2^ 44.12%), (−3.5 ng/mL, 95% CI −6.02 to −1.01, *p* = 0.006, I^2^ 77.24%, respectively). A trend of lower hemoglobin was observed in sarcopenic dialysis patients (−0.2 g/dL, 95% CI: −0.5 to 0.001, *p* = 0.06, I^2^ 87.61%) (Table 3).

### 3.6. Association between Sarcopenia and Clinical Outcomes

#### 3.6.1. All-Cause Mortality in Sarcopenia (LMM Plus LMS) Patients

Sarcopenia defined by combined criteria was significantly associated with higher mortality risk in both the unadjusted (OR 2.79, 95% CI 2.07 to 3.77; *p* < 0.001, I^2^ 27.2%) and adjusted analyses (OR 1.83, 95% CI 1.44 to 2.39; *p* < 0.001, I^2^ 40.6%). (Table 4, Figure 3). Subgroup analyses by using covariates including age, sex, comorbidity, nutritional indices, and inflammatory markers in multivariate logistic regression models were explored.

#### 3.6.2. All-Cause Mortality in Individual Components of the Diagnostic Criteria of Sarcopenia (LMM and LMS)

Five studies [23,31,40,50,57] (1447 patients) reported the association of LMM and LMS separately on mortality outcomes. It was found that LMM and LMS were independently related to increased mortality risk in dialysis patients (OR 1.71; 95% CI (1.20 to 2.44), *p* = 0.003, I^2^ 0%, OR 2.15 (95% CI 1.51 to 3.07), *p* < 0.001, I^2^ 16.7%), respectively (Table 4).

#### 3.6.3. Cardiovascular Events

CV events were reported in two studies [23,37] (175 patients) and were included in the meta-analysis. According to adjusted OR, sarcopenia was significantly associated with increased CV events in dialysis patients (adjusted OR 3.80, 95%CI: 1.79 to 8.09, *p* = 0.001, I^2^ 0%) (Table 4).

#### 3.6.4. Hospitalization

Two studies [25,50] followed dialysis patients to assess the impact of sarcopenia on hospitalization outcomes. The first study [25] included 126 chronic HD patients aged 63.2 ± 13 years with 3-year follow-up. The authors reported no significant difference in the incidence of hospitalization between sarcopenic and non-sarcopenic patients without absolute reported number. In contrast, the second study [50] followed 170 patients on maintenance HD for 3 years. The risk of hospitalization was significantly higher in sarcopenic patients with a crude RR of 1.80 (95% CI 1.35–2.41, *p* < 0.001) and a fully adjusted RR (adjusted for age, gender, dialysis vintage, and DM) of 2.07 (95%CI 1.48–2.88, *p* < 0.001).

#### 3.6.5. Dependency

Only one study [29] estimated the association of sarcopenia with dependency in activities of daily living (ADL). This study was performed in a total of 238 patients with maintenance HD with an average age of 60.9 ± 13.2 years. Univariate analysis showed that the presence of sarcopenia was significantly associated with dependency in both basic ADL (OR 2.69, 95% CI 1.39–5.20, *p* = 0.003) and instrumental ADL (OR 3.33, 95% CI 1.94–5.72, *p* < 0.001). However, after adjusting for clinical covariates, the OR for neither basic ADL nor instrumental ADL was significant (*p* > 0.05).

#### 3.6.6. Frailty

Only one study [21] reported the association of sarcopenia with frailty based on the Clinical Frailty Scale (CFS). This study evaluated 119 PD patients with the mean age of 66.8 ± 13.2 years and the mean follow-up period of 589.2 days. According to the multivariate logistic regression model, after adjusting for age, sex, BMI, nPNA, and Charlson comorbidity index (CCI), sarcopenia was significantly correlated with frailty (adjusted OR 12.2, 95% CI 2.27–65.5, *p* = 0.003).

#### 3.6.7. Investigations of Heterogeneity

Table 5 details the results of subgroup analyses comparing all-cause mortality between sarcopenic and non-sarcopenic dialysis patients as stratified by dialysis modalities (HD or PD), race (Asian or non-Asian), time of muscle mass measurement (pre-HD, intra-HD, or post-HD), tool of muscle mass measurement (DXA, BIS, BIA, or CC), follow-up duration (less or more than 2 years), components of diagnostic criteria (low muscle mass or low muscle strength), and five adjustment variables (age, sex, comorbidity, nutritional indices, and inflammatory markers). In brief, there was a significant association between dialysis patients with sarcopenia and all-cause mortality relative to non-sarcopenic dialysis patients across all subgroups and adjustment variables as mentioned above. However, there was significant heterogeneity based on the Q-test *p*-value and I^2^ index in some subgroups of the included studies, including Asian ethnicity and tools of muscle mass assessment by BIA.

#### 3.6.8. Meta-Regression Model

The meta-regression modeling, adjusted for sample size, was applied to mean ages, mean serum albumin, mean dialysis vintage, and crude prevalence rates from all studies, as these generated more data points. The modeling revealed that the prevalence of sarcopenia was not influenced by age (*p* = 0.63), serum albumin (*p* = 0.64), or dialysis vintage (*p* = 0.59).

#### 3.6.9. Assessment of Publication Bias

The funnel plot for the outcome of both sarcopenic prevalence and all-cause mortality in the studies included in the meta-analysis was asymmetrical (Appendix A). According to publication bias on all-cause mortality outcome, the sensitivity analysis was conveyed to identify the most influential study on the pooled adjusted odds ratio. We found that the pooled adjusted OR of all-cause mortality depended on a single study by Kim et al. [23] conducted in patients who had a high burden of co-morbid disease, especially cardiovascular disease. After excluding this study, the all-cause mortality outcome did not have a significant publication bias (*p* = 0.21) (Appendix A).

## 4. Discussion

ESKD patients undergoing KRT are at high risk of sarcopenia owing to the trend toward increased dialysis among the aging population incorporated with several ESKD-related sarcopenic factors. From this systematic review and meta-analysis, it was found that sarcopenia was highly prevalent among dialysis patients (overall pooled prevalence 25.6%) and widely varying, ranged from 1.5% to 68% (Figure 1), mainly depending on the applied diagnostic criteria, tools and time of muscle assessment, dialysis modalities, and abnormal fluid status in dialysis patients. Furthermore, this condition was also associated with adverse clinical outcomes, especially CV events and all-cause mortality. Dialyzed patients with sarcopenia were associated with increased risk of CV events (adjusted OR 3.80) and death (adjusted OR 1.83) compared with non-sarcopenic individuals (Table 4).

Our systematic review provided a worldwide epidemiologic representation of sarcopenic prevalence among dialysis patients, reporting the lowest absolute prevalence in the USA (15.4%), Australia (17.9%), increasing through South America (20.4%) and Asia (27.9%) and reaching the highest levels in Europe (29.1%) (Table 2). According to the diagnostic criteria from different guidelines [2,4,10,11,12,13,14,15], the highest prevalence was found in the Asian Working Group for Sarcopenia (AWGS) 2019 (36.9%) (Table 2). On the contrary, Liu X. et al. [60] illustrated that the prevalence of sarcopenia was the highest (57.1%) using EWGSOP 2010 criteria among 4500 Chinese participants. In this regard, the cutoff points of skeletal muscle mass index (SMI) and handgrip strength (HGS) are higher than the other five criteria (AWGS 2014, AWGS 2019, IWGS, FNIH, and EWGSOP 2019). However, the authors emphasized that applying the diagnostic criteria to different racial populations might be improper.

In ESKD patients undergoing KRT, both HD and PD, the frequency of sarcopenia is even greater than among subjects with normal renal function [61] because uremic-induced anorexia, metabolic disorder, and hormonal derangements inhibit muscle synthesis and accelerate muscle wasting [62]. Regarding dialysis modalities, our study found that PD patients seemed to have a significantly lower prevalence of sarcopenia than maintenance HD patients (Table 2). First of all, ESKD patients who chose PD rather than HD modality seemed to have a better health status as a result of younger age, less comorbidity and greater physical independence [63]. Moreover, the influence of each KRT modality on sarcopenia might be related to several issues including daily routine and the patient’s level of physical activity [64]. Most PD patients spend most of their time out of the healthcare center. They have more available time during the day, especially patients treated with nocturnal intermittent peritoneal dialysis (NIPD) mode, due to requiring only one or two dialysate exchanges per day at nighttime. On the other hand, HD patients must spend at least four hours, twice or thrice a week, with limited activity during transit to the HD center and during the HD session. In addition, after the HD session some HD patients may experience troublesome symptoms related to the HD procedure, such as fatigue, dizziness, cramping, etc. Therefore, on HD day, HD patients are often at increased need of rest and consequently favor a more sedentary lifestyle.

Another issue related to KRT modality is the assessment timing. This meta-analysis revealed that sarcopenia tended to be under-diagnosed by pre-HD compared with post-HD measurement of muscle mass (21.5% pre-HD to 27.8% post-HD), but with no statistical significance (*p* for interaction = 0.21) (Table 2). Similarly, a previous study [26] illustrated that post-dialysis measurement of muscle mass provided greater reliability. Although pre-dialysis measurement is more convenient, it certainly causes an error of muscle mass overestimation from a relatively high amount of water-contained muscle [65,66]. In addition, Yilmaz et al. [67] also revealed that over-hydration was statistically more frequent in PD than in post-HD patients (30.3% vs. 11.6%, *p* = 0.043). This factor may lead to the underestimation of the prevalence rate of sarcopenia in PD patients. Besides muscle mass assessment, measurement of muscle strength was also affected. Pinto et al. [68] demonstrated the dramatic reduction in HGS after HD compared with before HD sessions. The results showed a significant increase in the number of patients with HGS below the 30th percentile (44.9% before HD to 55.1% after HD; *p* < 0.01), particularly as it was related to post-dialysis fatigue caused by rapid removal of water and solutes during the procedure.

The last issue between HD and PD patients is that there is a strong body of evidence suggesting that PD better preserves residual kidney function (RKF) when compared with conventional HD [69]. The benefit of well-preserved RKF factor is greater elimination of protein-bound uremic toxins (PBUTs). Alcalde-Estévez. et al. [70] showed that PBUTs (combination of indoxyl sulphate and p-cresol) impaired the skeletal muscular regeneration process by inhibiting myoblast proliferation, reducing myogenic differentiation, and promoting muscular fibrosis, even at low concentrations, in a uremic rat model. From all the above factors including limited physical activity, the assessment timing, HD-related factors, and less-preserved RKF, the higher prevalence of muscle wasting and weakness in HD patients compared with PD patients was illustrated in our study (Table 5).

Diabetes mellitus (DM), one of the major health burdens and causes of KRT initiation worldwide [71], was shown to be negatively affecting various aspects of muscle health through impairments in protein metabolism, vascular and mitochondrial dysfunction by several different mechanisms, including inflammation, insulin resistance, advanced glycation end-product accumulation, and increased oxidative stress [72]. Mori. et al. [24] demonstrated that among 308 patients undergoing HD, DM as a comorbid disease was significantly associated with a higher rate in sarcopenic patients than non-sarcopenia individuals (41% vs. 27%, *p* = 0.015). Moreover, the presence of DM was an independent contributor to sarcopenia and an independent predictor of all-cause mortality in this population. In contrast, we found that diabetic patients were not significantly associated with greater risk of sarcopenia relative to non-diabetic individuals (OR 1.20, 95% CI 0.72 to 1.57, *p* = 0.18). This finding may be potentially explained by the better glycemic control in advanced CKD progress to ESKD requiring KRT from spontaneous resolution of hyperglycemia and normalization of glycated hemoglobin (HbA1C) levels designed, known as burn-out diabetes phenomenon [73]. In addition, DM is only one of the several ESKD-related factors in the development of sarcopenia. It is possible that the impact of DM might be overlaid by the other factors, including advanced age, long dialysis vintage, malnourishment, and CKD-Mineral and Bone Disorder (CKD-MBD) (Table 3).

Cardiovascular disease (CVD) is a highly common complication and the first cause of death in patients undergoing dialysis. Moreover, mortality due to CVD in this population is twenty times higher than in the general population [74]. Our meta-analysis demonstrated that sarcopenia in dialysis patients was one of the most important predictors of CV events as well as mortality outcome, and this was independent of study design, population, sex, continent, dialysis method, sarcopenia definition, and study quality (Table 5).

Regarding its components of diagnostic criteria, our meta-analysis illustrated that both LMM and LMS were independently related to increased mortality risk in dialysis patients. However, the magnitude of associated risks tended to be greater in the LMS component compared with the LMM component (Table 4). Several cohort studies [25,31,40,50,57] showed that dynapenia (i.e., LMS) could be superior to LMM for better prediction of adverse clinical outcomes, including quality of life, hospitalization, CV events, and all-cause mortality in ESKD patients. Consistent with these findings, Goodpaster et al. postulated that during the muscle wasting process, muscle strength decline occurs at a faster rate than muscle mass loss [75]. The proposed mechanism is that initial muscle weakness leads to decreased muscle function, diminished physical activity, and sometimes ends up with immobilization, as well as muscle disuse atrophy. Thus, decreased muscle mass and muscle strength are likely to be both the cause and the sequelae of each other. However, muscle mass in dialyzed patients is not the only determinant of muscle strength, but other factors such as muscle relaxation and fiber muscle atrophy can also explain LMS [76]. Indeed, the fulfilled diagnostic criteria of sarcopenia should be regarded as a late stage of muscle wasting since LMS alone (probable sarcopenia) is enough to trigger assessment of cause and provide early effective therapeutic intervention [2].

Although there are two previous meta-analyses on this topic (one that included 30 studies [11] and another that included 50 studies [77]), one of these reports [11] included several studies that defined sarcopenia as the presence of low muscle mass alone (or probable sarcopenia). In addition, not all studies used the validated sarcopenia diagnostic criteria. Thus, it could not be specified whether the prevalence and clinical outcomes exactly occurred from sarcopenia. In another previous report [77], the overall prevalence of sarcopenia was not explored. Moreover, the subgroup analyses according to dialysis modalities were limited due to a very small number of PD patients. Lastly, none of these reports explored the association of sarcopenia with CV events, one of the most important adverse clinical outcomes in dialysis patients.

The strength of the present study is that this is the first systematic review and meta-analysis of observational studies that demonstrates the worldwide pooled prevalence of sarcopenia and explores an association between sarcopenia and adverse clinical outcomes, especially cardiovascular events and mortality, among ESKD patients on dialysis. We included reports that performed multivariable analyses to account for potential confounders of these associations. In addition, the search included all studies published up to March 2022. More recent studies, using the EWGSOP 2019 as well as the AWGS that updated its sarcopenia definition and its cut-off points in March 2020, have been included. This might impact the recently updated overall prevalence of all classifications and subgroups by cut-off points. Despite these strengths, some limitations are worth mentioning. Firstly, our systematic review was limited to observational studies, and in the absence of randomized controlled trials, the cause-and-effect relation between sarcopenia and clinical outcomes remains speculative. Secondly, it is important to note the heterogeneity of the included studies due to the variety of population, the applied diagnostic criteria of sarcopenia, tools and time of muscle assessment, and dialysis modalities. Lastly, some clinical outcomes were reported by a small number of trials and more sample size is required to show a significant association.

## 5. Conclusions

In conclusion, our systematic review revealed that sarcopenia was highly prevalent among dialysis patients and demonstrated to be an important predictor of cardiovascular events and mortality. Additionally, both low muscle mass and low muscle strength were independently related to increased mortality risk. Of note, these findings could increase awareness of musculoskeletal health and encourage nephrologists who treat patients at risk of sarcopenia, leading to early detection and prompt implementation of beneficial therapeutic strategies. To optimize individual therapeutic approach and alleviate this burden of disease in dialysis patients, more future intervention research is needed.

## Figures and Tables

**Figure 1 nutrients-14-04077-f001:**
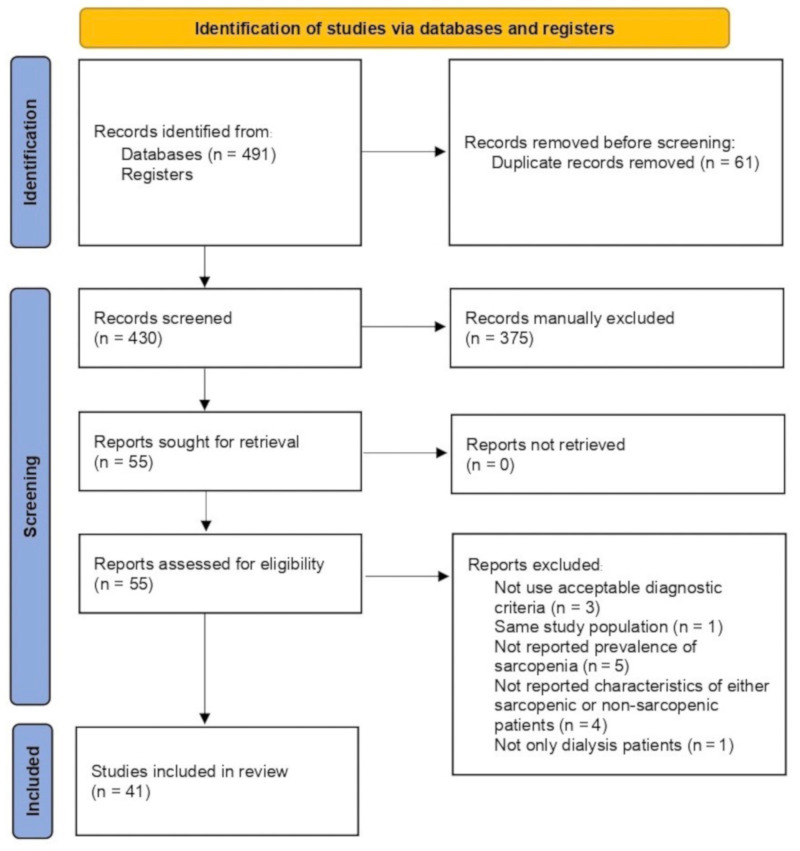
PRISMA 2020 flow diagram.

**Figure 2 nutrients-14-04077-f002:**
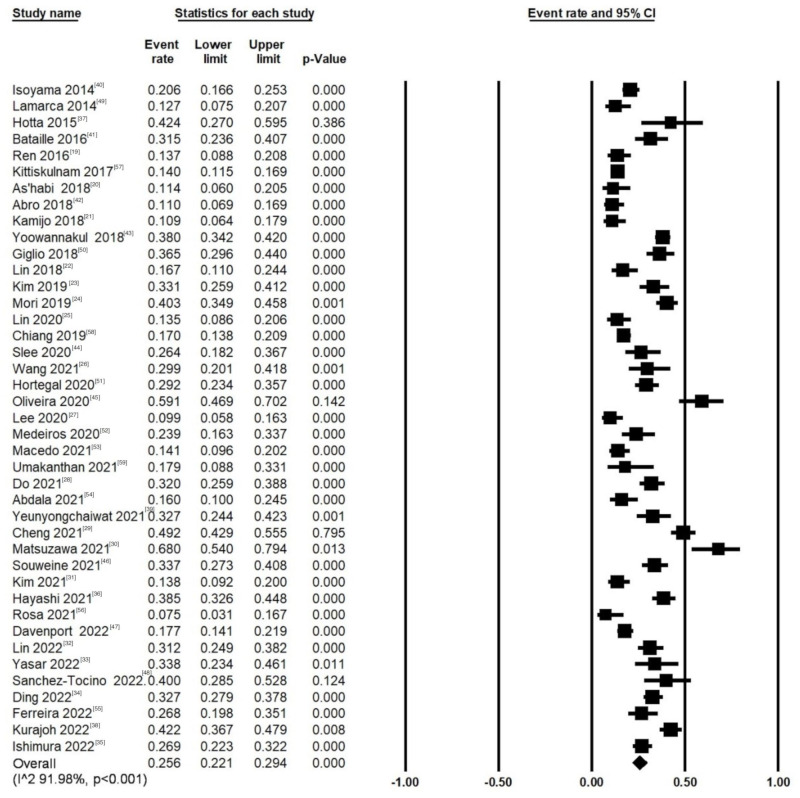
Forest plot visualizing the varying sarcopenic prevalence in dialysis patients reported as event rate for each study included publication in the meta-analysis [19,20,21,22,23,24,25,26,27,28,29,30,31,32,33,34,35,36,37,38,39,40,41,42,43,44,45,46,47,48,49,50,51,52,53,54,55,56,57,58,59].

**Figure 3 nutrients-14-04077-f003:**
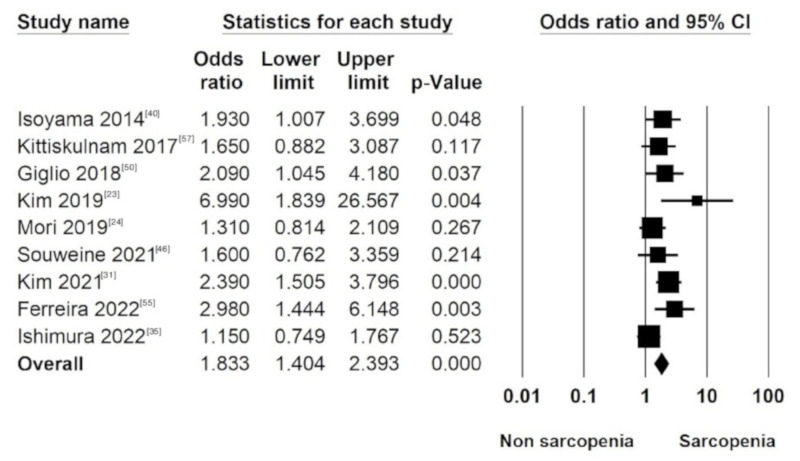
Forest plot displaying the pooled adjusted odds ratio for all-cause mortality among sarcopenic relative to non-sarcopenic dialysis patients [23,24,31,35,40,46,50,55,57].

**Table 1 nutrients-14-04077-t001:** Characteristics of the studies included in the systematic review.

Author	Year of Publication	Country	Design	No. of Patients	Mean Age (Year)	Men (%)	DM (%)	Dialysis Vintage (Month)	Mode of KRT	Operational Sarcopenia Criteria	Muscle Mass Instrument	Time of Muscle Mass Measurement	Muscle Strength Instrument	Physical Performance	F/U Time (Year)	Study Quality
Isoyama N.	2014	Sweden	Prospective cohort	330	53	62	31	NA	HD	EWGSOP 2010	DXA	Post-HD	HGS	NA	2.42	Good (7)
Lamarca F.	2014	Brazil	Cross-sectional	102	70.7	73.5	34	27	HD	EWGSOP 2010	BIA	NA	HGS	NA	NA	Good (7)
Hotta C.	2015	Japan	Cross-sectional	33	67.6	60.6	24.2	51.5	HD	EWGSOP 2010	BIA	NA	HGS, KEMS, OLST	GS	NA	Satisfactory (6)
Bataille S.	2016	France	Cross-sectional	111	77.5	58.6	52.3	35.4	HD	EWGSOP 2010	BIA	Intra-HD	HGS	NA	NA	Good (7)
Ren H.	2016	China	Cross-sectional	131	49.4	61.1	7.6	71.3	HD	EWGSOP 2010	BIA	Pre-HD	HGS	NA	NA	Good (7)
Kittiskulnam P.	2017	US	Prospective cohort	645	56.7	58.6	43.9	33.6	HD	EWGSOP 2010 + FNIH	BIS	Post-HD	HGS	GS	1.9	Good (7)
As’habi A.	2018	Iran	Cross-sectional	79	NA	44	38	NA	PD	EWGSOP 2010 + AWGS 2014	BIA	Dry abdomen	HGS	GS	NA	Good (7)
Abro A.	2018	UK	Cross-sectional	155	63	61.3	37.4	9	PD	FNIH, AWGS 2014, EWGSOP 2010	BIA	Dry abdomen	HGS	NA	NA	Good (7)
Kamijo Y.	2018	Japan	Prospective cohort	119	66.8	70.6	21	128.4	PD	AWGS 2014	BIA	NA	HGS	GS	1.61	Good (8)
Yoowannakul S.	2018	UK	Cross-sectional	600	66.3	62.2	46	30.9	HD	AWGS 2014, EWGSOP 2010, FNIH	BIA	Post-HD	HGS	NA	NA	Good (7)
Giglio J.	2018	Brazil	Prospective observational	170	70.6	65.3	62.4	34.8	HD	EWGSOP 2010	DXA	Intra-HD	HGS	NA	3	Good (7)
Lin Y.	2018	Taiwan	Cross-sectional	120	63.3	52.5	36.7	56.5	HD	EWGSOP 2010	BIA	NA	HGS	GS	NA	Good (7)
Kim J.	2019	Korea	Prospective observational	142	59.8	57	47.2	50.2	HD	EWGSOP 2010	BIA	Post-HD	HGS	NA	4.3	Good (7)
Mori K.	2019	Japan	Retrospective observational	308	58.06	60.1	32.8	77.3	HD	AWGS 2014	DXA	Post-HD	HGS	NA	6.33	Good (8)
Chiang J.	2019	US	Prospective cohort	440	56.1	100	41.1	32.4	HD	EWGSOP 2010 + FNIH	BIS	Pre-HD	HGS	NA	1	Good (7)
Lin Y.	2020	Taiwan	Prospective cohort	126	63.2	51.6	38.9	55.4	HD	EWGSOP 2010	BIA	Post-HD	HGS	GS	3	Good (8)
Slee A.	2020	UK	Cross-sectional	87	65.9	72.4	NA	61.7	HD	EWGSOP 2010, FNIH	BIA	Post-HD	HGS	NA	NA	Good (7)
Hortegal EVF.	2020	Brazil	Cross-sectional	209	51.9	59.3	35.8	NA	HD	EWGSOP 2019	DXA	Post-HD	HGS	GS	NA	Good (7)
Oliveira E.	2020	Spain	Cross-sectional	66	53.15	43.9	NA	NA	Mixed	EWGSOP 2010	BIA	NA	HGS	TUG	NA	Good (7)
Lee H.	2020	Korea	Cross-sectional	131	66.2	54.2	67.9	61.3	HD	AWGS 2014	BIA	Post-HD	HGS	NA	NA	Good (7)
Medeiros M.	2020	Brazil	Cross-sectional	92	63.3	63	44.5	NA	HD	EWGSOP 2010	BIA	Post-HD	HGS	NA	NA	Good (7)
Wang M.	2021	China	Cross-sectional	87	66.6	70.1	40.2	42.5	HD	AWSG 2014	BIA	Pre-HD, Post-HD	HGS	GS	NA	Good (7)
Macedo C.	2021	Brazil	Prospective observational	170	70.6	65.3	37.7	NA	HD	EWGSOP 2019	BIA	Post-HD	HGS	NA	3	Good (7)
Umakanthan J.	2021	Australia	Cross-sectional	39	69	72	31	37.4	Mixed	EWGSOP 2010	BIS	Pre-HD, random (PD)	HGS	NA	NA	Good (7)
Do J.	2021	Korea	Cross-sectional	200	55.5	57	49.5	57.8	PD	AWGS 2014	DXA	Dry abdomen	HGS	NA	NA	Good (7)
Abdala R.	2021	Argentina	Cross-sectional	100	55.7	60	NA	50.8	HD	EWGSOP 2019	DXA	Post-HD	HGS	GS, SST	NA	Good (7)
Yuenyongchaiwat K.	2021	Thai	Cross-sectional	104	59.7	51.9	37.5	70.3	HD	AWGS2019	BIA	NA	HGS	GS	NA	Good (7)
Cheng D.	2021	China	Cross-sectional	238	60.9	67.6	40.8	30.6	HD	AWGS 2019	BIA	Post-HD	HGS	GS	NA	Good (7)
Matsuzawa	2021	Japan	Cross-sectional	58	77.5	62.1	44.8	38.5	HD	AWGS 2019	BIA	Post-HD	HGS	GS	NA	Good (7)
Souweine J.	2021	France	Prospective cohort	187	65.3	65	15.5	67.2	HD	Other	BIA	Post-HD	HGS	VS	1.98	Good (8)
Kim C.	2021	Korea	Prospective observational	160	55.1	68.1	53.1	21.8	PD	Other	BIS	NA	HGS	NA	2	Good (8)
Hayashi H.	2021	Japan	Retrospective observational	244	66.6	70.5	41.4	134.7	HD	AWGS 2019	DXA	NA	HGS	GS	NA	Good (7)
Rosa CSC.	2021	Brazil	Cross-sectional	67	54.6	64.2	46.3	15.8	HD	AWGS 2019, EWGSOP 2010, EWGSOP 2019, FNIH	DXA, BIA	Non-HD day	HGS	NA	NA	Good (7)
Davenport A.	2022	UK	Retrospective observational	368	60.9	61	39.7	14.2	PD	AWGS 2019 + EWGSOP 2019	BIA	Dry abdomen	HGS	NA	NA	Good (7)
Lin Y.	2022	Taiwan	Cross-sectional	186	57.5	46.2	40.3	45	PD	AWGS 2019, EWGSOP 2019, FNIH, IWGS	BIA	NA	HGS	GS	NA	Good (7)
Yasar E.	2022	Turkey	Cross-sectional	65	44.9	56.9	20	132	Mixed	EWGSOP 2019	BIA	Pre-HD Dry abdomen (PD)	HGS	NA	NA	Good (7)
Sanchez-Tocino M.	2022	Spain	Prospective observational	60	81.9	68	NA	49.9	HD	EWGSOP 2019	BIA	Intra-HD	HGS	GS, TUG, SPPB	NA	Good (7)
Ding Y.	2022	China	Cross-sectional	346	58.2	61.1	28	52.7	HD	AWGS 2019	BIA	Post-HD	HGS	GS	NA	Good (7)
Ferreira M.	2022	Brazil	Prospective cohort	127	NA	56.6	30.7	30.7	HD	EWGSOP 2010, EWGSOP 2019	CC	Post-HD	HGS	GS	1.96	Good (8)
Kurajoh M.	2022	Japan	Cross-sectional	296	68	68.6	57.8	78	HD	AWGS 2019	DXA	NA	HGS	CST	NA	Good (7)
Ishimura E.	2022	Japan	Retrospective cohort	308	58	60.1	32.8	49.2	HD	AWGS 2019	DXA	Post-HD	HGS	NA	6.3	Good (8)

Abbreviations: AWGS, Asian Working Group for Sarcopenia; BIA, bioimpedance analysis; BIS, bioimpedance spectroscopy; CC, calf circumference; CST, Chair Stand Test; DM, diabetes mellitus; DXA, Dual-Energy X-ray Absorptiometry; EWGSOP, European Working Group on Sarcopenia in Older People; FNIH, Foundation for the National Institutes of Health; F/U, follow up; GS, gait speed; HD, hemodialysis; HGS, handgrip strength; IWGS, International Working Group on Sarcopenia; KEMS, knee extensor muscle strength; KRT, kidney replacement therapy; Mixed, peritoneal dialysis and hemodialysis; NA, not available; OLST, One-Leg Standing Test; PD, peritoneal dialysis; SPPB, Short Physical Performance Battery; SST, Sit Stand Test; TUG, Timed Up and Go; UK, United Kingdom; US, United States; VS, Voorrips score.

**Table 2 nutrients-14-04077-t002:** Subgroup analyses examining the pooled prevalence of sarcopenia in different variables among dialysis patients.

Subgroup Analysis	No. Studies	No. Patients	Heterogeneity		Model	Pooled Prevalence % (IQR)
			*p*-Value	I^2^		
Diagnostic criteria						
AWGS 2014	8	1667	<0.001	91.49%	Random	22% (15.6–30.0%)
AWGS 2019	9	1839	<0.001	88.67%	Random	36.9% (30.2–44.2%)
EWGSOP 2010	17	2498	<0.001	89.40%	Random	24.4% (19.3–30.4%)
EWGSOP 2019	8	948	<0.001	82.04%	Random	24.1% (18.0–31.4%)
FNIH	5	1095	<0.001	82.04%	Random	20% (13.8–28.0%)
IWGS	1	186	1.00	0%	Random	34.9% (28.4–42.1%)
Mixed †	4	1530	0.243	28.10%	Random	15.7% (13.6–18.1%)
Other ‡	2	347	<0.001	94.30%	Random	22.4% (8.5–47.3%)
Tools of muscle mass measurement						
BIA	26	3935	<0.001	91.46%	Random	26.2% (21.5–31.5%)
BIS	4	1282	0.499	0%	Random	15.2% (13.3–17.3%)
CC	1	127	1.00	0%	Random	26.8% (19.8–35.1%)
DXA	10	2232	<0.001	88.54%	Random	29.2% (23.7–35.3%)
Dialysis modalities						
HD	31	6139	<0.001	92.11%	Random	26.8% (22.8–31.2%)
PD	7	1267	<0.001	88.69%	Random	17.5% (11.9–24.8%)
Mixed	3	170	<0.001	88.62%	Random	36.2% (17.2–60.8%)
Time of muscle mass measurement						
Intra-HD	6	635	<0.001	82.52%	Random	25.8% (18.1–35.3%)
Post-HD	16	3967	<0.001	93.97%	Random	27.8% (22.2–34.3%)
Pre-HD	5	2232	0.002	76.53%	Random	21.5% (15.0–29.8%)
Continents						
Asia	21	3453	<0.001	90.86%	Random	27.9% (23.0–33.4%)
Australia	1	39	1.00	0%	Random	17.9% (8.8–33.1%)
Europe	9	1962	<0.001	92.66%	Random	29.1% (21.5–38.0%)
North America	2	1085	0.171	46.60%	Random	15.4% (12.6–18.6%)
South America	8	1037	<0.001	84.51%	Random	20.4% (14.7–27.5%)

Abbreviation: AWGS, Asian Working Group for Sarcopenia; BIA, bioelectrical impedance analysis; BIS, Bioimpedance spectroscopy; CC, calf circumference; DXA, dual-energy X-ray absorptiometry; EWGSOP, European Working Group on Sarcopenia in Older People; FNIH, Foundation for the National Institutes of Health; HD, hemodialysis; IWGS, International Working Group on Sarcopenia; Mixed, peritoneal dialysis and hemodialysis; PD, peritoneal dialysis. † EWGSOP 2010 + FNIH, EWGSOP 2010 + AWGS 2014, EWGSOP 2010 + FNIH, AWGS 2019 + EWGSOP 2019. ‡. One study (Kim et al.) used LTI below the tenth percentile of a reference population plus HGS below 28.9 kg in males and below 16.8 kg in females according to cutoff values in Korean. Another study (Souweine et al.) used muscle strength and mass below the median of both maximal voluntary force (MVF) and creatinine index (CI).

**Table 3 nutrients-14-04077-t003:** Meta-analysis of weighted mean differences in demographic characteristics and laboratory parameters among sarcopenic versus non-sarcopenic dialysis patients.

Variables	No. Studies	No. Patients	Heterogeneity		Model	Meta-Analysis	
			*p*-Value	I^2^		WMD (95%CI)	*p*-Value
Age	19	3504	<0.001	73.80	Random	8.81 (7.10, 10.53)	<0.001
BMI	15	2523	<0.001	82.66	Random	−2.87 (−3.62, −2.12)	<0.001
Dialysis vintage	18	2845	0.329	10.50	Random	5.56 (0.88, 10.24)	0.020
Serum albumin	19	3429	0.003	54.15	Random	−0.13 (−0.18, −0.09)	<0.001
Serum phosphate	11	1976	0.063	43.06	Random	−0.62 (−0.81, −0.44)	<0.001
Serum PTH	8	1154	0.038	52.89	Random	−48.39 (−94.60, −2.18)	0.040
Serum creatinine	12	2240	0.050	44.12	Random	−1.63 (−1.95, −1.30)	<0.001
Serum CRP	16	2665	<0.001	92.01	Random	1.307 (0.07, 2.54)	0.038
Serum 25-OH vitamin D	6	642	0.001	77.24	Random	−3.514 (−6.02, −1.01)	0.006
Hemoglobin	13	2371	<0.001	87.61	Random	−0.25 (−0.50, 0.01)	0.055
Kt/V	9	1508	0.001	71.09	Random	0.11 (0.06, 0.17)	<0.001
FTI	3	396	<0.001	92.01	Random	−3.51 (−6.02, −1.01)	0.006

Abbreviation: BMI, body mass index; CRP, C-reactive protein; FTI, fat tissue index; PTH, parathyroid hormone; WMD, weight mean difference.

**Table 4 nutrients-14-04077-t004:** The association of sarcopenia, low muscle mass (LMM) and low muscle strength (LMS) with all-cause mortality and CV events among dialysis patients.

First Author (Year of Publication)	Sarcopenia		Low Muscle Mass (LMM)		Low Muscle Strength (LMS)		Adjustment Variables
	Unadjusted Odd Ratio (95% CI)	Adjusted Odd Ratio (95% CI)	Unadjusted Odd Ratio (95% CI)	Adjusted Odd Ratio (95% CI)	Unadjusted Odd Ratio (95% CI)	Adjusted Odd Ratio (95% CI)	
All-cause mortality							
Isoyama N. (2014)		1.93 (1.01–3.71)		1.23 (0.56–2.67)		1.98 (1.01–3.87)	Age, sex, diabetes, CVD, cholesterol, Hb, GFR and hs CRP
Kittiskulnam P. (2017)	2.46 (1.48–4.09)	1.65 (0.88–3.08)	2.2 (1.39–3.46)	1.7 (0.94–3.05)	2.42 (1.55–3.77)	1.68 (1.01–2.79)	Age, sex, race, DM, CHF, CAD and albumin
Giglio J. (2018)	2.02 (1.14–3.57)	2.09 (1.05–4.2)	1.49 (0.79–2.82)	1.6 (0.73–3.53)	2.03 (1.09–3.79)	1.84 (0.92–3.68)	Age, gender, dialysis vintage and DM
Kim J. (2019)		6.99 (1.84–26.58)		2.77 (1.10–6.97)		5.65 (1.99–16.04)	Age, gender, BMI, KT/V, albumin, DM, dialysis vintage, hs CRP, previous history of CAD and CVD
Mori K. (2019)		1.31 (0.81–2.1)					Age, HD vintage, gender, BMI, DM, Hb, albumin, CRP
Souweine J. (2021)	3.0 (1.5–6.0)	1.6 (0.76–3.35)					Age, sex, LTI, albumin, hs CRP, serum bicarbonates, dialysis vintage and Charlson score
Kim C. (2021)		2.39 (1.51–3.81)		2.1 (1.12–8.29)		3.61 (1.14–11.41)	Age, gender, BMI, dialysis duration, DM and albumin
Ferreira M. (2022)		2.98 (1.44–6.13)					Age, DM, COPD, CHF, HIV infection and HCV infection
Ishimura E. (2022)		1.15 (0.75–1.77)					NA
	Pooled 2.79 (2.07–3.77)	Pooled 1.83 (1.40–2.39)		Pooled 1.71 (1.20–2.44)		Pooled 2.15 (1.51–3.07)	
Cardiovascular events							
Kim J. (2019)		4.33 (1.51–12.43)		3.01 (1.09–8.29)		4.09 (1.26–13.29)	Age, gender, BMI, KT/V, albumin, DM, dialysis vintage, hs CRP, previous history of CAD and CVD
Hayashi H. (2021)		3.31 (1.12–9.76)					NA
		Pooled 3.80 (1.79–8.09)					

Abbreviation: BMI, body mass index; CAD, coronary artery disease; CHF, congestive heart failure; COPD, chronic obstructive pulmonary disease; CVD, cardiovascular disease; DM, diabetes mellitus; GFR, glomerular filtration rate; Hb, hemoglobin; HCV, hepatis C virus; hs CRP, high sensitive C-reactive protein; HIV, human immunodeficiency virus.

**Table 5 nutrients-14-04077-t005:** Subgroup analyses examining the association between sarcopenia and all-cause mortality in dialysis patients.

Subgroup Analyses	No. of Studies	No. of Patients	Pooled Adjusted Odds Ratio (95% CI)	*p*-Values	Assessment of Heterogeneity	
					I2 Index	*p*-Value
Dialysis modalities						
PD	1	160	2.39 (1.51–3.80)	<0.001	0%	1.00
HD	8	2152	1.75 (1.31–2.33)	<0.001	38.24%	0.125
Race						
Asian	4	918	1.81 (1.07–3.06)	0.027	71.49%	0.015
Non-Asian	5	1394	1.98 (1.46–2.68)	<0.001	0%	0.755
Time of muscle mass measurement						
Intra-HD	1	170	2.09 (1.05–4.18)	0.037	0%	1.00
Post-HD	7	2142	1.72 (1.25–2.38)	<0.001	0%	0.755
Tools of muscle mass measurement						
DXA	4	1053	1.43 (1.09–1.87)	0.010	2.78%	0.379
BIS	2	803	2.10 (1.45–3.04)	<0.001	0%	0.351
BIA	2	329	3.00 (0.72–12.52)	0.132	72.07%	0.058
CC	1	127	2.98 (1.44–6.15)	0.003	0%	1.00
Study follow-up time						
≤2 years	4	1401	2.16 (1.41–3.30)	<0.001	33.19%	0.213
>2 years	5	911	1.64 (1.18–2.28)	0.003	39.54%	0.158
Adjusted demographic characteristics						
Yes	7	NA	1.90 (1.41, 2.56)	<0.001	24.04%	0.246
No	1	NA	2.39 (1.51, 3.80)	<0.001	0%	1.00
Adjusted co-morbidities						
Yes	5	NA	2.36 (1.74, 3.21)	<0.001	7.26%	0.365
No	3	NA	1.52 (1.08, 2.14)	0.016	0%	0.634
Adjusted nutrition						
Yes	5	NA	1.74 (1.21, 2.49)	0.003	28.81%	0.23
No	3	NA	2.43 (1.73, 3.41)	<0.001	0%	0.782
Adjusted inflammatory markers						
Yes	4	NA	1.84 (1.13, 3.01)	0.015	46.6%	0.132
No	4	NA	2.23 (1.65, 3.00)	<0.001	0%	0.654
Adjusted anemia						
Yes	2	NA	1.50 (1.02, 2.20)	0.039	0%	0.346
No	6	NA	2.23 (1.69, 2.95)	<0.001	4.0%	0.391

Abbreviations: BIA, bioimpedance analysis; BIS, bioimpedance spectroscopy; CC, calf circumference; DXA, Dual-Energy X-ray Absorptiometry; HD, hemodialysis; NA, not available; PD, peritoneal dialysis.

## Data Availability

The datasets generated during and/or analyzed during the current study are available from the corresponding author on reasonable request.

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
