# Peer review of "Prevalence of Sarcopenia and Its Impact on Cardiovascular Events and Mortality among Dialysis Patients: A Systematic Review and Meta-Analysis"

_nutrients, 2022, doi:10.3390/nu14194077_

Round 1

Reviewer 1 Report

Dear Authors, 

The bias, accuracy, comprehensiveness and precision with which the research was done make this kind of research valid.  Meta-analyses require a special quality assessment and therefore their results are a challenge.  This study was conducted to assess the prevalence of sarcopenia, the diagnosis of which has different criteria.  In addition, the question is how the information from the source was selected, i.e. whether the articles were selected only by searching the database.  Regarding the quality of the article, apart from the way of reporting, which is satisfactory, I do not find any special contribution because the information obtained does not bring anything new.

The paper published in the journal J Cahexi Sarcopenia Muscl 2022 Feb;13(1):145-158. doi: 10.1002/jcsm.12890. by the author Xiaoyu Shu under the title "Diagnosis, prevalence and mortality of sarcopenia in dialysis patients: a systematic review and metaanalysis" has the same methodology.

Author Response

Response to Reviewer 1 Comments

Point 1:The bias, accuracy, comprehensiveness and precision with which the research was done make this kind of research valid.  Meta-analyses require a special quality assessment and therefore their results are a challenge.  This study was conducted to assess the prevalence of sarcopenia, the diagnosis of which has different criteria.  In addition, the question is how the information from the source was selected, i.e. whether the articles were selected only by searching the database.  Regarding the quality of the article, apart from the way of reporting, which is satisfactory, I do not find any special contribution because the information obtained does not bring anything new. The paper published in the journal J Cahexi Sarcopenia Muscl 2022 Feb;13(1):145-158. doi: 10.1002/jcsm.12890. by the author Xiaoyu Shu under the title "Diagnosis, prevalence and mortality of sarcopenia in dialysis patients: a systematic review and metaanalysis" has the same methodology.

Response 1:  Thank you very much for your comment. We totally agree with the limitation of meta-analysis of cohort studies. Therefore, we explored the study quality and reported the results in the supplementary table. In addition, we also explored the known confounding factors in adjusted multivariate models and subgroup analysis for testing the precision of the results. The overall results seem to be consistency.

        In term of previous meta-analysis (J Cahexi Sarcopenia Muscl 2022), we emphasize and give more specific PICO in our review (as shown in the table below). First, our inclusion criteria included only the trials that use true definition of sarcopenia (defined by low muscle mass and low muscle strength) and did not include probable sarcopenia (low muscle mass alone). Thus, the prevalence and clinical outcomes exactly occurred from sarcopenia. Second, our review focused more about the association of sarcopenia with adverse clinical outcomes, including mortality, CV events, frailty, dependency and hospitalization. Third, our review provided more recent trials (published after 2019). Using the EWGSOP 2019 as well as the AWGS 2019 that updated its sarcopenia definition and its cut-off points in March 2020 have been included. This might impact the recently updated overall prevalence of all classifications and subgroups by cut-off points. Finally, we also explored the known confounding factors in both adjusted multivariate models and subgroup analysis for testing the precision and accuracy of the results.

Previous trial

Our trial

Definition of sarcopenia

Total 30 studies

: LMM alone (probable sarcopenia) 8 studies

: LMM + LMS (sarcopenia) 22 studies

Total 41 studies

LMM + LMS (sarcopenia)

: 41 studies

Clinical outcomes

Mortality

Mortality

Cardiovascular events

Frailty

Dependency

Hospitalization

No. of studies that published after 2019

8

26

Reviewer 2 Report

The authors reviewed the prevalence of sarcopenia and the impact on CV events and mortality in dialysis patients more in detail when compared with the previous review papers. So, I have no additional comment for this review.

Author Response

Response 1:  Thank you very much for your comment.

Reviewer 3 Report

This paper reviewed the prevalence of sarcopenia and its impact on cardiovascular events and mortality among dialysis patients by systematic review of observational studies. Although this study was not a systematic review and meta-analysis of RCT as recommended by the GRADE guideline, it is overall well-written and well-analyzed.

Major concerns and questions for authors:

1.     The most significant concern is that, as described above, this systematic review and meta-analysis were not performed regarding RCTs. Therefore, in addition to the absence of cause-and-effect relation, the impact of several confounding could not be excluded (especially the impact of unmeasured covariates). Additionally, the multivariate adjustments used in these observational studies must be different from one another.

2.     In page 17, lines 450–463 – Regarding the discussion upon the impact of KRT modalities on sarcopenia, the effect of baseline characteristics who choose each modality could not be excluded. Generally, the patients who choose PD rather than HD are younger, have less comorbidity, and physically and socially independent or well-assisted. Additionally, PD is difficult to control fluid status compared to HD, which could lead to the overestimation of SMI assessed with DXA or BIA and could lead to underestimation of the prevalence of sarcopenia.

3.     In the section 3.6.5 Malnutrition, not the impact of nutritional status on sarcopenia, but the impact of sarcopenia on malnutrition was assessed. Please consider to delete or modify this section.

Minor concerns and questions for authors:

1.     Were there any studies excluding younger individuals, especially those who were under 65 (as defined in some criteria of sarcopenia) ?

2.     Page 17, lines 464–467 – In the Discussion section, this part seemed different: the sarcopenia tended not “over-diagnosed,” but “under-diagnosed” in pre-HD compared to post-HD. Actually, in the Results section, the prevalence of sarcopenia was 21.5% and 27.8% in pre-HD and post-HD, respectively.

3.     The dialysis vintage of HD and PD has quite different meaning. Please clarify the vintage of each modality, as well as that of total participants. Additionally, were there any patients on hybrid therapy with HD + PD?

Author Response

Response to Reviewer 3 Comments
      This paper reviewed the prevalence of sarcopenia and its impact on cardiovascular events and mortality among dialysis patients by systematic review of observational studies. Although this study was not a systematic review and meta-analysis of RCT as recommended by the GRADE guideline, it is overall well-written and well-analyzed.

Major concerns and questions for authors:

Point 1: The most significant concern is that, as described above, this systematic review and meta-analysis were not performed regarding RCTs. Therefore, in addition to the absence of cause-and-effect relation, the impact of several confounding could not be excluded (especially the impact of unmeasured covariates). Additionally, the multivariate adjustments used in these observational studies must be different from one another.

Response 1: Thank you very much for your suggestion. We totally understand about your concern. and highlight this issue in the limitation part of manuscript. According to most of the studies of sarcopenia in dialysis patients were limited to observational studies. Therefore, we explored the known confounding factors in adjusted multivariate models (table 4) and subgroup analysis (table 5) for testing the precision of the results. All the adjustment variables were categorized into 5 domains (e.g. demographic characteristics, co-morbidities, nutritional status, inflammatory markers and anemia) and other possible confounding risk factors, including dialysis modalities, race, time and tool of muscle mass measurement and study follow-up time were included in this subgroup analysis. Eventually, it was shown that there were positive correlations between sarcopenia and mortality outcome across all subgroups. 

Recently, there was few robust evidence (RCT) that explored the effects of therapeutic intervention (e.g. exercise, nutritional therapy) on the clinical outcomes that related with sarcopenia. This meta-analysis would like to emphasize the association between sarcopenia and adverse clinical outcomes to encourage the future therapeutic research to intervene this kind of disease burden.

Point 2: In page 17, lines 450–463 – Regarding the discussion upon the impact of KRT modalities on sarcopenia, the effect of baseline characteristics who choose each modality could not be excluded. Generally, the patients who choose PD rather than HD are younger, have less comorbidity, and physically and socially independent or well-assisted. Additionally, PD is difficult to control fluid status compared to HD, which could lead to the overestimation of SMI assessed with DXA or BIA and could lead to underestimation of the prevalence of sarcopenia.

Response 2: Thank you very much for your suggestion. We have clarified more about this issue in this discussion part (line 444-446, 463-466).

Point 3: In the section 3.6.5 Malnutrition, not the impact of nutritional status on sarcopenia, but the impact of sarcopenia on malnutrition was assessed. Please consider to delete or modify this section.

Response 3: Thank you very much for your suggestion. We have deleted this section in the revised manuscript.

Minor concerns and questions for authors:

Point 4: Were there any studies excluding younger individuals, especially those who were under 65 (as defined in some criteria of sarcopenia)?

Response 4: Yes, there are 4 studies that included only elderly dialysis patients (age > 60 years) as shown in the table below

No.

Author (published year)

N

Inclusion of patients’ age

Prevalence

2

Lamarca F. (2014)

102

Elderly > 60 years

12.7%

11

Giglio J. (2018)

170

Elderly > 60 years

36.5%

23

Macedo C. (2021)

170

Elderly > 60 years

14%

37

Sanchez-Tocino M. (2022)

60

Age between 75-95 years

40%

Point 5: Page 17, lines 464–467 – In the Discussion section, this part seemed different: the sarcopenia tended not “over-diagnosed,” but “under-diagnosed” in pre-HD compared to post-HD. Actually, in the Results section, the prevalence of sarcopenia was 21.5% and 27.8% in pre-HD and post-HD, respectively.

Response 5: Thank you very much for your comment. We apologize for our mistake. We have already corrected this error in the revised version.

Point 6: The dialysis vintage of HD and PD has quite different meaning. Please clarify the vintage of each modality, as well as that of total participants. Additionally, were there any patients on hybrid therapy with HD + PD?

Response 6: Thank you very much for your comment. The mean dialysis vintage of HD and PD, as well as total of participants (from the reported trials) are shown in the table below. In addition, there were no patients on hybrid therapy from the included trial.

Modalities

Mean dialysis vintage (month)

No. of participants

Hemodialysis

52.9

5,093

Peritoneal dialysis

58.3

1,253
